# Female adolescents living with HIV telling their story through 'my story' book in Malawi: A visual methodology innovation

**Gertrude Mwalabu** [1] *, **Ida Mbendera**[2], **Pammla Petrucka**[3], **Violet Manjanja**[4]

1 Department of Adult Health Nursing, Kamuzu College of Nursing, University of Malawi, Lilongwe, Malawi,
2 Department of Public Sector Management, Jubilee University, Lilongwe, Malawi, 3 College of Nursing,
University of Saskatchewan, Saskatoon, Canada, 4 Department of Midwifery, Kamuzu College of Nursing,
University of Malawi, Lilongwe, Malawi

* mwalabugees2004@kcn.unima.mw

## Abstract

journal.pone.0257126

SOUTH AFRICA

**Data Availability Statement:** All relevant data are
within the paper and its Supporting Information
files.

### Introduction

Female adolescents living with Human Immunodeficiency Virus are a unique population fac-
ing lifelong challenges in reproductive and sexual health ranging from relational, social and
legal-ethical considerations. While HIV prevalence of young females is higher than the
males (4.9% versus 1%), evidence show that these adolescents in Malawi initiate sex as
early as 15 years mostly with adult partners. Yet, young people are frequently reported to be
reluctant to discuss sexual matters and demonstrate avoidance of direct questions on sex-
ual issues during interviews. It is critical therefore that researchers invoke youth-friendly
approaches to address these complex issues and enable these vulnerable individuals to
articulate and advocate for their preferred futures.

### Methods

This study used an innovative visual qualitative approach known as 'my story' book which
combined image selection and sentence completion exercises to enable adolescents living
with HIV to share their stories through interviews. The study involved 14 cases comprising
of 14 female adolescents aged 15 to 19, 14 caregivers and 14 health providers working at
HIV multidisciplinary centres in Malawi.

### Results

The 'my story' book enabled in-depth exploration of the experiences and issues faced by
this vulnerable population. The use of images and sentence completion exercise enabled
the researcher to appreciate the type of interactions (or lack of) on sexual issues and status
disclosure to significant others including sexual partners. Three themes were revealed,
included creating meanings, revealing confirmatory and/or complementary evidence and
enabling intergenerational research.

**Funding:** This study was funded by the Malawi Government through the University of Malawi in the form of academic scholarships and the University of Malawi through the Faculty of Nursing in the form of academic scholarships used to support data collection and analysis and the writing of this paper.

**Competing interests:** The authors have declared that no competing interests exist.

**Abbreviations:** AIDS, Acquired Immune Deficiency Syndrome; ARVs, Antiretroviral; HIV, Human Immunodeficiency Virus; SRH, Sexual and Reproductive Health.

## Conclusions

The 'my story' book captured the life experiences and needs of the female adolescents. The book assisted the participants in constructing, communicating, and controlling the articulation of their stories. It positioned female adolescents as a diverse group of social agents able to construct and reflect on their social worlds and real-life issues. This approach was highly effective in creating a youth-friendly context for reflection and revelation regarding sensitive cultural and social issues faced by this group. As such, the 'my story' book could be used in one-on-one counselling or provide data to inform intervention development.

## Introduction

Globally, the number of young people living with perinatally acquired HIV continues to rise, mainly in Africa, including Malawi. Between 2015–2016 HIV prevalence among young people aged 15–24 years was significantly higher among females (4.9%) than their male counterparts (1%) in Malawi [1]. Evidence suggests that young women are particularly vulnerable to sexual abuse and this situation exists in the sub-Saharan Africa context including Malawi [2, 3]. Many HIV positive females in Malawi initiate sex by the age of 15, mostly with older or non-cohabiting sexual partners for material support and survival, and these encounters are overwhelmingly unprotected [1, 4]. This pattern often results in poor sexual and reproductive health (SRH) outcomes, including unplanned pregnancies and early childbearing. HIV services in Malawi provide support for the young women's HIV-related clinical needs but still face unprecedented challenges of meeting the SRH needs of this growing population as they grow into adulthood. There is little specific provision for young people transitioning from child to adult within the health systems in HIV Management centres [2, 4]. The services offered in the centres include counselling and teen club meetings which are conducted once every month. The main activities include: medical care (including condom distribution), group discussions on sexual and reproductive issues and games.

However, evidence suggests that health professionals encounter challenges in addressing young women's distinctive SRH needs and it becomes a challenge for the majority of young people to access contraceptives offered by adults [2, 4]. This is attributed to service providers being ill-equipped to discuss sexual issues with young people.

While challenges on discussing sexual issues with young people are well documented elsewhere [4–6], likewise in Malawi premarital sex and parent-child discussion on sexual issues is culturally unacceptable for the fear of influencing young people to initiate sex too early or because sexual discussions are customarily held in secret within families [4, 7]. Yet, 98% of girls aged 15–19 are aware of modern methods of contraception and only about one-third of sexually active unmarried young women use contraceptives [8]. Reasons cited for lack or limited uptake of the services include fear of being seen as not conforming to religious, cultural and social norms, negative attitude of service providers, inability to negotiate safer sex with partners and misconceptions [2, 4, 7]. A growing body of qualitative studies has also shown that inferior socio-economic status, lack of bargaining power in sexual relationships coupled with gender power inequality (in terms of experience, authority, and control over sexual activity) affect the female adolescents' ability to take informed decisions particularly on safer sex and contraception [2, 4, 5, 7, 8]. This becomes more complicated particularly in Africa where young people are frequently reported being reluctant to discuss sexual matters and

demonstrate avoidance of direct questions on sexual issues during interviews particularly those living with HIV [4, 5]. All these consistently point to a need for more information, open communication, friendly and flexible services for young people [9], and calls for innovative data collection tools to enable young people in articulating the realities of their lives including sexual and reproductive issues with adult researchers.

Kellett and Robinson as cited by Fraser indicate that it can be methodologically difficult for an adult researcher to find techniques to listen to the young people's voices in research [6]. They argued that the gap in generations between the adults and young people creates differences in viewpoints, experiences and power, and it can become difficult to bridge this gap. In response to this concern, Samuels stressed the value of techniques for "bridging the culturally distinct worlds of the researcher (an adult) and the researched (young people)" [10]. To address this concern, the incorporation of visual forms of expression has become common in qualitative research over the past two decades, with participant-employed photography being most prevalent. For instance, the evolution of the 'memory book', in the United Kingdom as an innovative method for biographical research dates as early as 2005 [11]. Memory books were created as a method to be used alongside interviews in a longitudinal qualitative study of young people's transitions to adulthood as critical tools for the understanding of identity, sources of documentation and resources for elaboration. The method assisted in facilitating the expression of a range of 'voices' and time frames which complicate cohesive narrative presentations of self. Visual methods such as photovoice have also been used in community-based studies and with individuals to explore their lived experiences, particularly because of their participatory nature [12]. With photovoice, researchers often aim to decrease power differentials in the research process and empower young research participants [13], which aligns with empowerment models that are becoming increasingly advocated by health professionals [14]. Photovoice study conducted in Canada with young adult women affected by serious illness, provides examples of participants' photographs to illustrate how participant-employed photography could enhance the depth of research data. In particular, the examples highlighted how the photographs enriched participants' verbal descriptions of their lived experiences, which generated a better understanding of their personal embodied realities. Guillemin and Drew [15] in their study further indicated that images have the ability to "express the unsayable" (p. 178). Images can therefore be "both a form of data and a conduit for the elicitation of interview data thereby revealing more and greater details than other methods alone would have generated" [16] (p. 239). However, in the Sub-Saharan region, including Malawi there is paucity of literature on use of photovoice in data collection, despite young people being reported to be reluctant to discuss sensitive issues including sexual behaviours and HIV [4, 5, 7]. Our paper therefore adds to this body of knowledge by focusing specifically on 'my story' book (a novel data collection tool with young people). In addition, it is the first innovative data collection method to particularly consider the global and Malawian contexts.

## Materials and methods

### Study design

A qualitative case study was conducted in Malawi with 14 female adolescents with perinatally-acquired HIV between January—November 2012. Each 'case' comprised a female adolescent (aged 15–19 years), a nominated caregiver, and a service provider.

### Settings

This study was conducted at three HIV Management Centres providing services to paediatrics and young people living with HIV (two urban centres and one rural centre). All the centres

offer comprehensive paediatric HIV care and support, including SRH, counselling, nutrition, entrepreneurship, and family planning services to young people. The rural facility is affiliated to the public facility which is the first specialist centre in Malawi for the care and support of people living with HIV [4]. By the time of data collection in 2012, the centre excluding its sites had registered over 20,000 HIV positive patients; out of which 387 (192 females, 195 males) were young people aged 12 to 19 years. The other facility is a non-governmental and non-profitable organization working in partnership with the Ministry of Health and manages 10% of all children commenced on antiretrovirals (ARVs) in Malawi, including young people. By 2012, the centre had a total of 2825 active patients, out of which 956 were young people: 635 were aged 10 to 14 years (325 females, 310 males), and 321 were aged 15 to 19 years (157 females, 164 males).

## Ethics approval and consent to participate

The University of Nottingham Research Ethics Committee in the United Kingdom (C 09 2011), and College of Medicine Research and Ethics Committee in Malawi (P.09/11/1124) approved all study protocols. Each participating specialized paediatric HIV centre gave written permission to conduct the study.

Verbal and written informed consents were obtained from the adolescents aged 18 and above and their caregivers. Assent was obtained from adolescents under age 18 in tandem with parental permission for their participation because young people are considered 'persons with diminished autonomy hence vulnerable to coercion or undue influence [17]. The informed consent offered the 'opportunity to hear their views of any kind of taken-for-granted issues regarding research with young people' [18], and the participants were recruited into the study with full disclosure. Pseudonyms invented by participants themselves (adolescents, caregivers and service providers) were used to ensure maximum anonymity and confidentiality.

## Recruitment and sampling

The adolescents were recruited through the multidisciplinary centres providing specialised paediatric HIV care and serving this study's target population. The healthcare professionals approached adolescents who met the inclusion criteria and their primary caregivers to ask them to participate in the study. Both the adolescents and their caregivers were approached on an individual basis if they expressed an interest in participating in the study. Adolescents who visited the centres on their own were given information sheets to share with their caregivers which invited them to participate in the study and bring feedback during the following visit. If the caregivers were unable to attend the centre, the lead author visited them to provide further explanation of the study and followed this visit up with a phone call or another visit. The young woman and her caregiver were then given an opportunity to identify a service provider who had been in constant contact with the young woman for at least six months to round out the 'triad' for the study.

14 participants were recruited through purposive sampling, five adolescents aged 15–19 were recruited from each urban HIV management centre, and four from the rural facility (two urban centres and one rural centre). The lead author's personal experience shaped the age of adolescents selected for the current study. Based on her own experience as an adult female and mother, the lead author felt that older adolescents had wider experience than younger adolescents, which was consistent with the existing literature [4, 19, 20]. All adolescents who met the selection criteria were invited to participate in the study by the service providers. The criteria included perinatal exposure to HIV, awareness of their HIV positive status, clinic attendance for a minimum of six months, and a cognitive capacity to complete 'my story' book.

**Table 1. Profile of participating female adolescents (as referred to on page 5).**

| Pseudonym | Residential area | Age | Type of caregiver | Educational level | Access to specialised HIV care | Sexual characteristics | | | | |
|---|---|---|---|---|---|---|---|---|---|---|
| | | | | | | Main source of financial & material Support | Sexual debut | No. of partners | Number of children | Age of child bearing |
| Alindine | Rural | 19 | Aunt | Tertiary | √ | Aunt | X | 1 | x | x |
| Chitsanzo | Rural | 19 | Sister | Secondary | √ | Sexual partners | 15 | 3 | 1 | 17 |
| Dalo | Urban | 19 | Mum & Dad | Primary | √ | Sexual partners & small business | 12 | unknown | 2 | 15 |
| Fatsani | Urban | 17 | Mum & Dad | Secondary | √ | Mum & Sexual partners | 14 | 4 | x | x |
| Gonjetso | Rural | 16 | Aunt | Secondary | √ | Aunt, teacher & service provider | X | x | x | x |
| Mwatitha | Urban | 18 | Aunt | Secondary | √ | Aunt & sexual partners | 12 | 3 | x | x |
| Nane | Urban | 19 | Dad | Tertiary | √ | Dad | 16 | 2 | x | x |
| Penina | Urban | 19 | Dad | Secondary | √ | Dad | X | 1 | x | x |
| Tamando | Urban | 17 | Mum & Dad | Secondary | √ | Sexual partner & small business | 12 | 1 | 1 | 15 |
| Tanyada | Urban | 16 | Mum & Dad | Primary | √ | Mum & Dad | X | x | x | x |
| Tawina | Rural | 18 | Uncle | Secondary | √ | Sexual partners & uncle | 14 | 5 | x | x |
| Ulemu | Urban | 18 | Aunt | Secondary | √ | Small business | X | x | x | x |
| Zaiwo | Urban | 19 | Mum | Primary | √ | Mum | 14 | 1 | 1 | 14 |
| Ziliwe | Urban | 18 | Aunt | Secondary | √ | Sexual partners | 15 | unknown | 1 | 17 |

Adolescents who were illiterate, pregnant, already had children, or were married were also included if they fulfilled the above criteria. Adolescents that were unaware of their exposure to HIV, and/or were either clinically unwell or too sick to complete 'my story' book were excluded. This exclusion was implemented to reduce any additional stress to such adolescents and to minimise the potential loss of informants due to critical illness prior to completion of the study. Out of the 20 adolescents who met the inclusion criteria, only 14 were able to assemble as 'complete cases', (i.e. each case comprised of an adolescent, her caregiver and a service provider). Six adolescents did not participate because their primary caregivers were not willing to participate in the study for fear of being associated with the adolescents' HIV status. Out of the 14 cases, ten were recruited from the urban setting and four from the rural centre (See Table 1). Only adolescents whose caregivers also participated in the study (for complete cases) were given a full explanation about the use of 'my story book and took part in the sentence completion exercise. All adolescents that participated in the current study gave written content to publication of the cases' details with pseudonyms invented by adolescents themselves.

## Data collection using 'my story' book

Data from female adolescents was collected through semi-structured interviews using the open-ended questions that formed the 'sentence completion exercise' in 'my story' book (See Fig 1). The concept of 'my story' book, loosely originated from the idea of 'memory books' and shares foundational elements of photo voice. A memory book is a tool used in memory work with individuals living with HIV. It is a diary of a typical day, which includes spaces for drawing [11, 21]. However, 'my story' book included a sentence completion exercise and researcher-generated images depicting different life experiences, events and issues potentially affecting lives of the female adolescents as they are growing up with HIV (See Figs 2–7) in

DEFINITIVE SENTENCE COMPLETION EXERCISE IN "MY STORY" BOOK

Major themes:

- Experiences of growing up with HIV infection
- Major needs/issues that impact on adolescent as they grow up to adult hood and adult care. May include:

    ✓ Sexual and reproductive health
    ✓ Disclosure of HIV status
    ✓ ART adherence
    ✓ Psychosocial support
    ✓ Future aspirations and priorities.
    ✓ Other medical issues

NB: Have a look at the pictures and choose those that best match your responses to the questions to be completed and use the space provided to write your response or create an image that suits your response best.

If you cannot read or write, the researcher will explain to you how you should just put stickers on images that best match your responses.

Experiences of growing up with HIV infection (Use yellow stickers for images depicting your responses)

| Questions | Response |
|---|---|
| Tell me your story about growing up with HIV (an adolescent will be asked to explain more about the response during interviews).<br><br>- What makes you say that? | |
| How do you feel about having HIV?<br><br>(Put stickers on images that depict your responses)<br><br>What do you think makes you feel like that? | |
| How does having HIV effect your:<br><br>- relationships with peers<br>- relationships with family members - future plans | |
| If the relationship is affected or not affected, what do you think are the possible reasons for that?<br><br>How about the reasons for the effects you have mentioned upon your future plans? | |

B. Major needs/issues as you grow up to adulthood and adult care (use green stickers for images depicting needs/issues)

| Questions | Response |
|---|---|
| What might be your major challenges or difficulties of living with HIV as you are growing up to adulthood (Put stickers on images that suit your major challenges or create images).<br><br>What are your reasons for choosing these images? | |
| Who do you like to talk to about your condition? (Put stickers on pictures that suit your preferred individuals).<br><br>Why talking to the people chosen?<br><br>Why not talking to the others about your condition? | |
| What do you hope for in the future? (Put stickers on pictures that suit your future desires or create images).<br><br>Do you think your condition influenced you in any way in regard to images chosen?<br><br>If so how?<br>If not, why not? | |
| What helps you to cope up with living with your condition? (Put stickers on pictures that depict on issues that help you to cope or create images)<br><br>How does that help you cope up with your condition? | |
| What do you think are the most important and realistic strategies in meeting your needs?<br><br>Why do you think so? | |

**Fig 1. Definitive sentence completion exercise in "my story" book.**

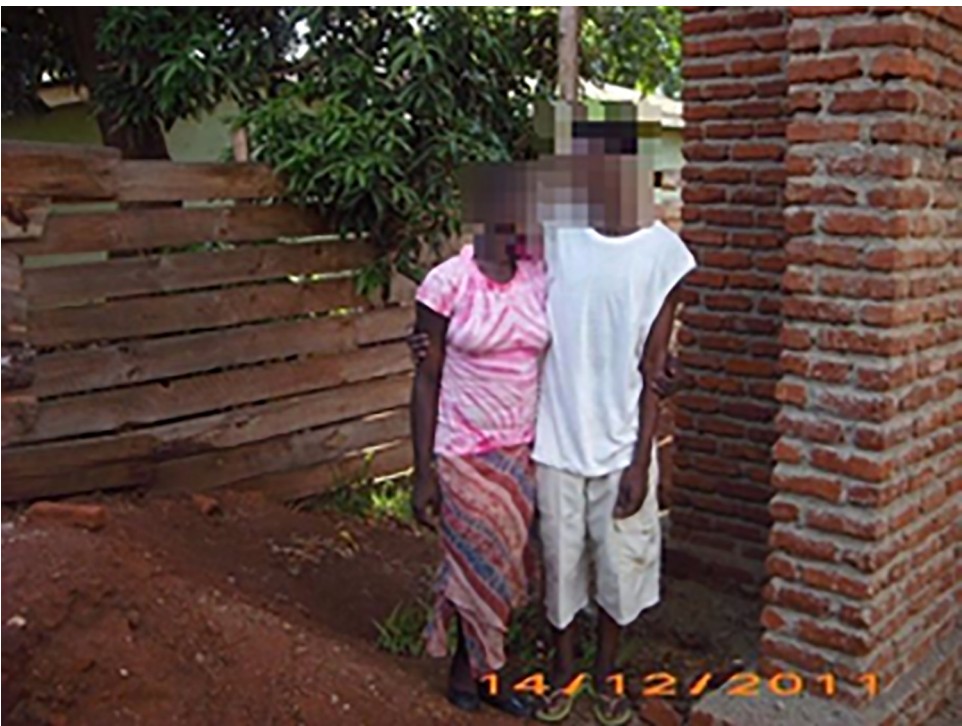

**Fig 2. Having a partner.**

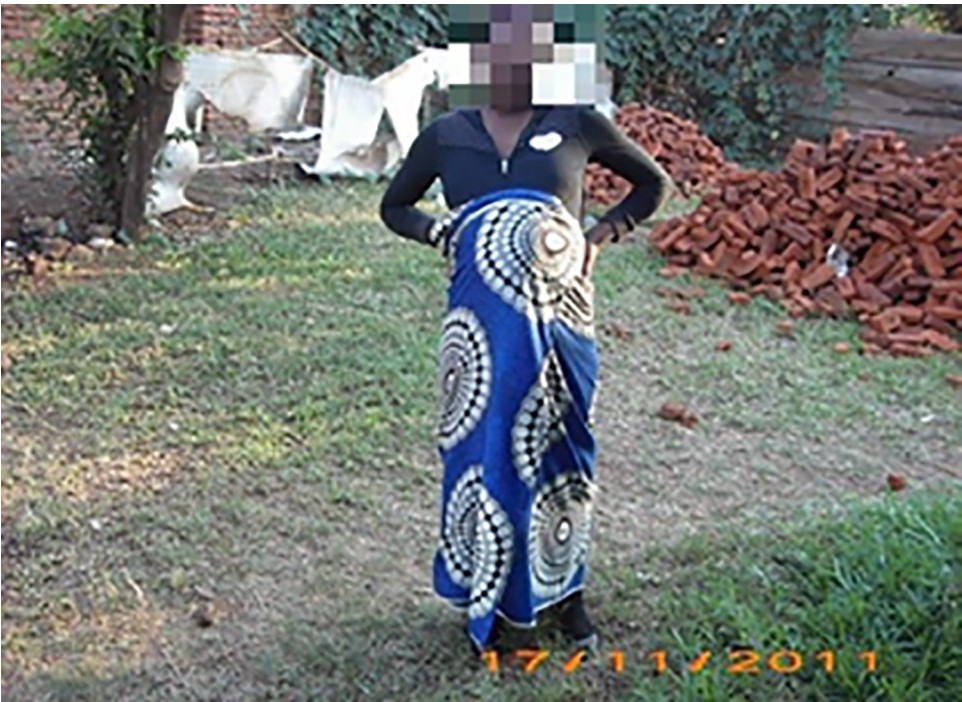

**Fig 3. Image of 'pregnancy'.**

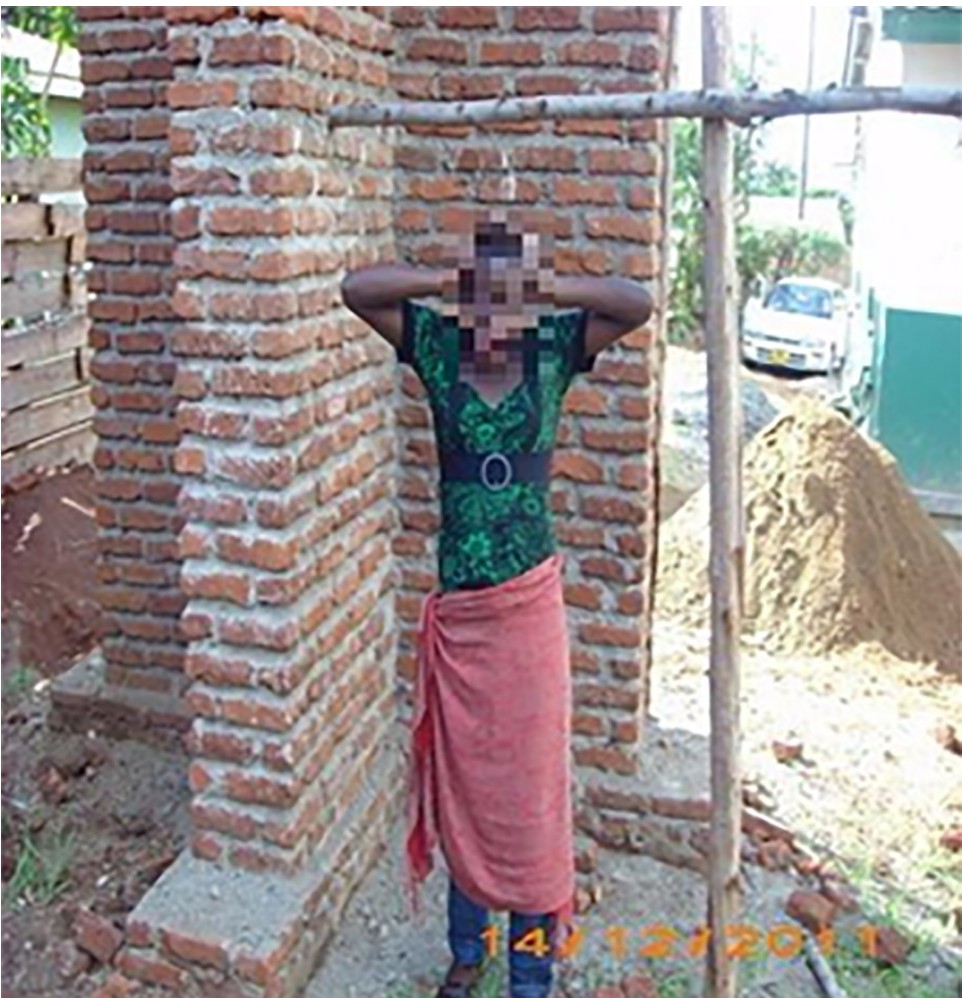

**Fig 4. Am depressed.**

English and Chichewa (Malawian language). As asserted by Teachman et al. [18] and Wills et al. [19], the inclusion of photographs helps generate thick description of data. Furthermore, central to 'my story' book were the explanations, interpretations, and meanings surrounding the chosen images.

The main themes of the 'my story' book which formed the interview guide included experiences of growing up with HIV infection, SRH, disclosure of HIV status, ART adherence, psychosocial support, future aspirations and priorities and other medical issues. The themes were derived from the literature, empirical data, research objectives, consultations with experts on the research topic, and an intent to challenge assumptions and taboos (both cultural and intergenerational) and also informed the choice of images [18, 20]. The consulted experts (six) were the medical and nursing specialists in Paediatric HIV and AIDS directly working with the young people in the HIV management centres. The use of literature and research objectives in the choice of images also assisted in reducing bias throughout the research. In this case, the researcher critically reflected on how and what kind of knowledge to produce, and how to relate the new knowledge to that already in existence.

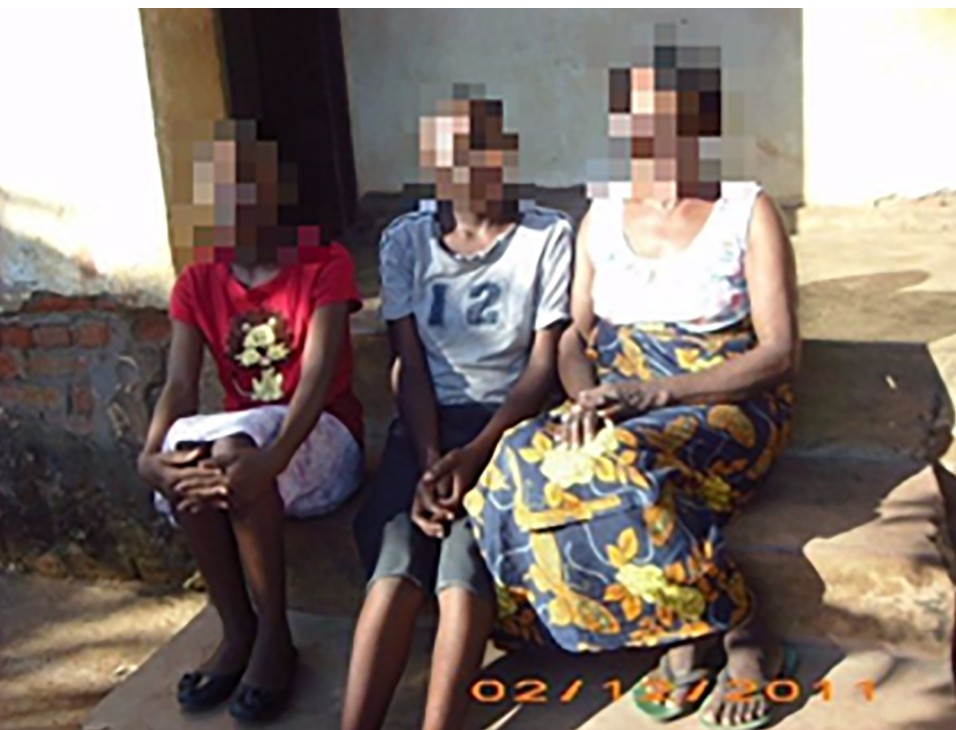

**Fig 5. Family members.**

The indicative images were extracted from the copyright free websites such as Wikimedia Commons. The definitive images were captured after pretesting the indicative images with female adolescents living with perinatally acquired HIV at a different health facility offering HIV services, to determine their applicability and relevance in the Malawian context (See Figs 2–7). Adolescents were invited to put stickers on images that best reflected their different emotions/experiences, major needs, and issues that impacted upon their lives, future aspirations and priorities. The plan was for adolescents to complete the 'sentence completion exercise' (See Fig 1) at their own time and pace within the scheduled period of two months. However, some young women (three) took more than two months (three to four months) to complete 'my story' book. Follow up visits (ranging from three to seven visits) to adolescents were conducted to establish a degree of trust in the research relationship and to enable adolescents to complete the 'my story' book. Out of 14, 13 adolescents filled all sections of 'my story' book (See Figs 1–7).

There was also provision of plain papers to allow adolescents to draw images or suggest topics for cases where the images were not applicable to their situations but none of the adolescents used the plain papers. The images were used as a catalyst to guide in-depth interviews with the researcher, as the adolescents were asked to explain their reasons for selecting particular images and the meaning(s) attached to them. With reference to the chosen images, an open-ended question such as *'can you tell me about your story of growing up with HIV?'* was posed to initiate the interviews with the adolescents. This question enabled the adolescents to recount details from any aspect of their life depending on the chosen images. Adolescents discussed those cultural aspects of their social world and social relations that they considered relevant, in accordance with the written/filled in reasons for choosing the images or responses in their 'sentence completion exercise'. Follow up questions included, *'How do you feel about*

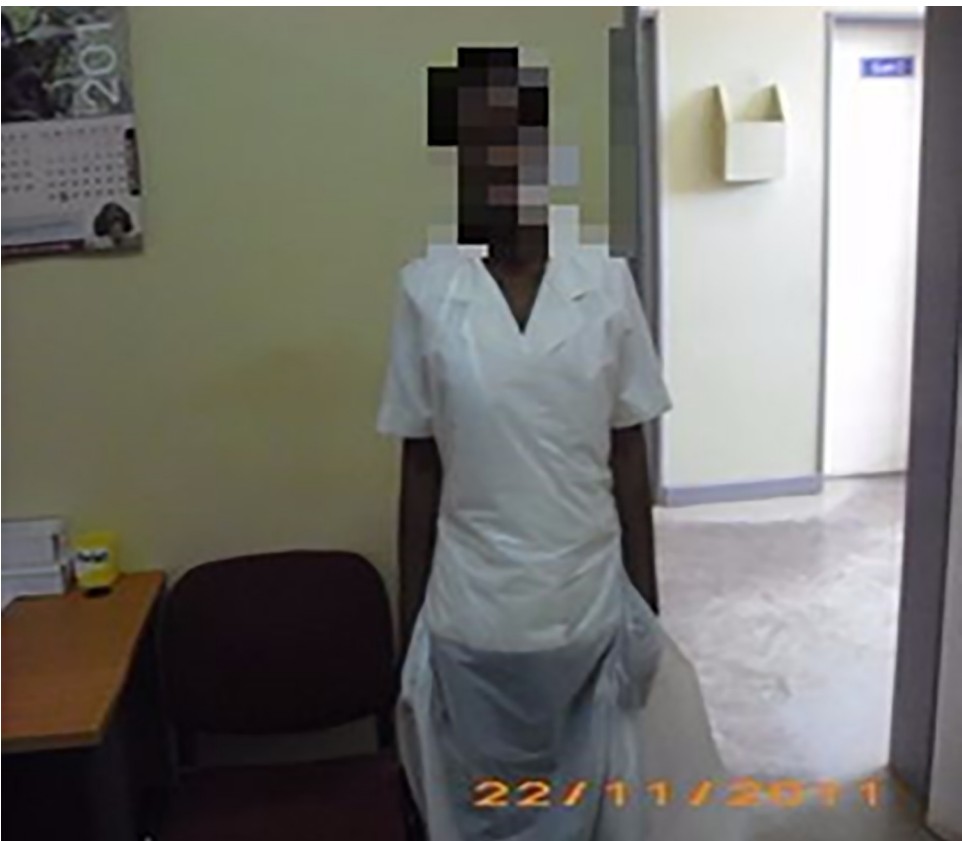

**Fig 6. Service providers.**

*having HIV*? *What do you think makes you feel like that*? *What might be your major challenges or difficulties of living with HIV as you are growing up to adulthood*? *What helps you to cope up with living with your condition*? The questions created the setting in which the adolescents were allowed to sketch the big-picture issues surrounding their lives. Thus, 'my story' book assisted young women to actively construct the story of their experiences, which communicated not only key events but also contexts, feelings, values, challenges and opinions as they were explaining the experiences in reference to the chosen images. Prompts and probes developed as the interviews progressed to encourage the respondents to think more deeply and facilitate openness for the complexity and uniqueness of individual experiences, challenges and perceived needs for young women.

The interviews were conducted by the lead author according to the participant's schedule, availability and preferences in terms of venue to minimise distress and ensure that participants felt at ease, particularly the young women. In situations where interviews were conducted at participants' homes, other family members were asked to stay at a distance, or sit outside of the house where possible, while the interview was in progress to ensure privacy and confidentiality. However, a 21year old caregiver had a repeated interview because she was distressed during the interviews as she recounted their relationship with an uncle (a husband to their aunt) who used to sexually abuse them, and was referred for support services within the centre.

Semi-structured interview guides were used to collect data with caregivers and service providers. All interviews lasted between 30 to 90 minutes and were recorded using an audio digital recorder. There were no incentives given apart from refreshments and transport refunds

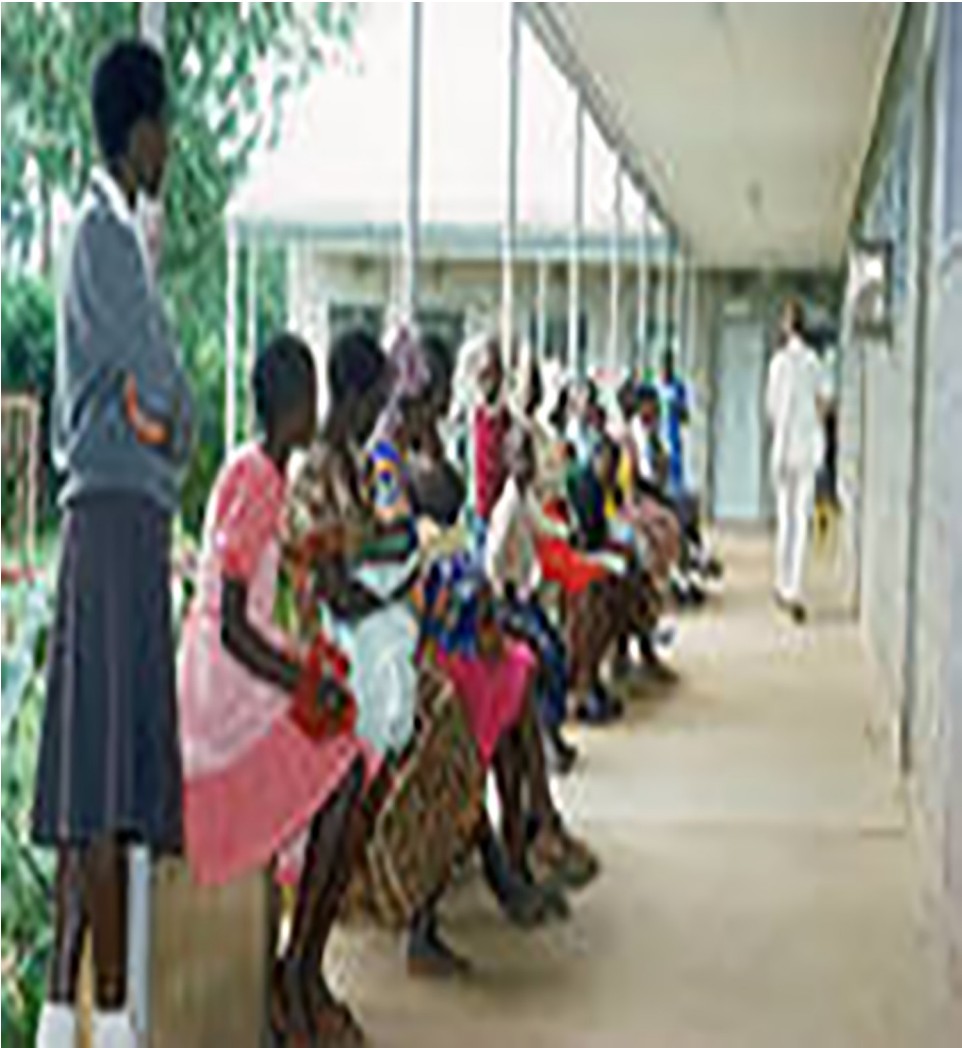

**Fig 7. Type and approach of health services.**

where necessary. Interviews conducted in Chichewa were mostly transcribed straight into English by the lead author herself, as a native speaker of Chichewa. Each transcript (the translated) was then audited through proof reading and spot checking during its development against the original audio recording by a lecturer in language and communication (bilingual expert), who is also content expert and an HIV and AIDS work place coordinator. After data analysis, the lead author returned all booklets to adolescents during adolescents' following visits to the clinic. Audio recording provided a complete description of participants' responses and comments in the context with which they were made, hence facilitating the interview process [22]. The lead author transcribed all the recorded interviews verbatim soon after each interview. The keeping of a thematic log during in-depth interviews, maintaining detailed researcher field notes, writing reflective remarks during data collection, ongoing reading of the literature and transcript readings were components of the research approaches that facilitated decision about data saturation and ongoing sampling [23].

## Data analysis

A multi-step retroductive process, including thematic analysis within and across cases, was done to produce contextually grounded, transferable findings. Retroduction is the process of "moving backwards from an observed event and asking, what must be true in order to make this event possible?" [24]. Easton states that, in 'retroductive analysis, events are explained by identifying mechanisms which are capable of producing the events' (p. 123). This iterative process of uncovering underlying structures and mechanisms was relevant to the topic under study. Thus, the retroductive analysis provided an alternative way of answering 'why' questions [25]. The interviews were analysed thematically using a constant comparison approach of different data sets [25]. Images were analyzed with respect to their content and the meanings assigned to them by participants during interviews.

Individual accounts within each 'case' were analysed. Codes, sub-themes, and themes were identified inductively from the individual accounts within the case then conclusions were drawn. The lead author engaged in inter-rater reliability activities to ensure credibility of the study findings. The lead author and her PhD research supervisors coded the same data set independently and achieved a high level of agreement on the codes. The outcome of the method ranged from 0 (no agreement at all) to 1 (agreement). We often used a unique code only once in a transcript. We would determine the value for each code in a transcript comparing all the three coders together and then combine the specific values into an average across the board. We considered reliability measured in aggregate for each transcript (total number of agreements among the coders).

Each transcript within a case (for all the 14 cases) incited potential reconsideration of the previously identified themes as commonalities and differences emerged. This enabled researchers view each case individually within its own context, and remain true to the case study approach [26]. Cross-case analysis was then undertaken to identify similarities, differences, relationships, and contradictions in the accounts of adolescents, caregivers, and service providers across the cases to draw conclusions (See S1 Dataset–Chapter 9 Study conclusion). The comprehensive analysis of the whole data set from multiple sources of evidence (images, sentence completion exercise) were linked to data from semi-structured interviews within the individual 'case', across the 'cases' to strengthen each case. The results related to the content are forthcoming.

Trustworthiness of the study findings was achieved through ensuring credibility and transferability of the study results. To enhance the rigour of the interpretive process, emerging themes were discussed with the research supervisors and the selected respondents. Participant validation exercises (through meetings) were conducted with the young women, caregivers, and service providers (i.e. separately for each category of participants) after data collection to also ensure the credibility of the results. This mechanism enabled the participants to comment on whether the lead author's tentative interpretations reflected their expressed experiences and enable further discussion about emerging issues and themes. During participant validation exercise, all participants affirmed that the lead author's interpretation of the data captured and expressed their personal experiences. In order to enhance the potential for transferability of the findings of the current study, a multiple case study approach was adopted. In addition, a diverse sample of young women, based on a range of demographic, socio-economic and HIV-related variables were selected to gain in-depth knowledge of a wider group. Furthermore, the comprehensive analysis of the whole data set from multiple sources of evidence coupled with the comparison of the findings within the individual 'case', across the 'cases' then with other findings in the existing literature likely allows transferability of the findings to young women growing up with perinatal HIV in a similar context.

## Results

### Participants demographic characteristics

14 adolescents completed the 'my story' book component of the study. Six reported living with their biological parent(s) (father/mother), three were married and living with their husbands, and five were double orphans living with either an aunt, an uncle, or a sister. Five adolescents were still attending school (four in secondary school and one in primary school), one was in college, one had completed studies at a tertiary level, and two were out of school (due to failed exams and/or lack of fees). Five participants had children. None of the adolescents was employed and the majority depended on their caregivers for survival and/or adopted other means of survival. This broad profile of the female adolescents in terms of age, parenthood, settings (rural/urban), literacy level, economic status, sexual behaviour, as well as marital and child bearing status contributed to strengthening of the study's capacity to compare and contrast their experiences, and identify commonalities and/or differences in their needs.

### Significance of 'my story' book

Through the sentence completion exercise, participants indicated reflective relevance of the chosen images to their sexual and reproductive experiences, needs, and challenges. In addition, the exercise probed the participants to express their experiences or views and the meanings they ascribed to their chosen images [9, 21]. This paper therefore highlights the methodological potentials that 'my story' book demonstrated under the following themes: i) creating meanings; ii) enabling an understanding of ambiguities and complex issues; and iii) gaining insight into inter-generational dissonance.

**Creating meanings.**   With the use of 'my story' book, adolescents were competent social actors. This was demonstrated during interviews as they were asked to elaborate or explain about their chosen images, and the meaning(s) they attached to them in relation to their sexual and reproductive needs. Most participants reported that accompanying loss of parental affection and care due to death of a parent(s), they experienced a diminished sense of belonging to family, peers, and society. Relationships with parents, caregivers, aunts, uncles, step-parents, siblings and significant others fundamentally influenced not only how one perceived one's self, but also perceptions of their value or worth to the social world. The strong attachment with a parent/caregiver or significant others was observed as an important ingredient to self-acceptance and positive self-image, as they were fundamental in building economic capital. However, in the case of Ziliwe, who chose images seen in Fig 2. 'Having a sexual partner' and Fig 3. Image of 'Pregnancy', indicated that losing both parents meant loss of love and adult support networks. As such, she engaged in intimate relationships to seek love, acceptance, and support. Ziliwe felt that her partner understood her challenges and was very supportive, which boosted her self-image as a young woman. She further reported making repeated efforts to negotiate condom use, but what became apparent was a common trajectory of events in which talking about condom use resulted in arguments and potential loss of the desperately needed support. Instead, she succumbed to her partner's demands and engaged in unprotected sexual encounters, which resulted in early pregnancy. She described her experience as follows:

> *I was passing through hard life at home, no-one cared for me; he gave me his ears, he supported me, and thus how I got connected to him; he accepted me and used to give me money which I used to buy food, and my necessities like body lotion, clothes which made me look like other girls. . .I felt loved, gave back in kind (sex); he could not accept using them (condoms) we argued and I just gave up and yielded to his demands, I became pregnant.* (Ziliwe, 18).

Similarly, Zaiwo explained the reasons for engaging in early sexual activities through rationale and meaning ascribed to an image of 'having a sexual partner'. Through the chosen image, Zaiwo was able to explain her early child bearing as attributed to lack of financial/material support which led to her school dropout, and early sexual activities to earn a living. In responding to how she felt that she is HIV positive, Zaiwo selected an image of 'Am angry with my illness' and proffered the explanation that she felt bitter because her parents passed on to her a shameful and secretive disease.

Through 'my story' book, young women reported encountering several challenges that influenced their SRH choices contrary to caregivers' and service providers' expectations. For instance, Tawina selected three images depicting use of contraceptives, taking medications (i.e., Anti-retrovirals [ARVs]), and type/approach of health services as she was responding to, *'what might be your major challenges or difficulties of living with HIV as you are growing up to adulthood?'*. In her accounts in the exercise and during interviews, Tawina elaborated on her choices that she could not disclose to her sexual partner that she was taking ARVs for fear of losing the support gained. The exercise enabled her to explain why she could not suggest use of condoms as the fear of being suspected by the boyfriend that she was HIV positive and also wanted to portray to service providers that she was not engaging in sexual activities as service providers advocated abstinence to young women living with HIV in order to reduce risk of transmitting the virus.

Young women's sexual health became more complicated as they sought love, acceptance and/or financial support. They reported being in a dilemma because their partners, upon whom they were materially dependent insisted on sex without condoms as a demonstration of love and for sexual pleasure, leading to early child bearing. Ziliwe, Chitsanzo, Tamando, Dalo and Tawina believed that apart from condoms, if the injectable contraceptives were readily available to them, their teenage pregnancies could have been prevented.

In contrast, the majority of service provider emphasised on abstinence and condom use, as the best options. Furthermore, Ms Kando, Ms Mbenu, Mrs Tayenda and Mr Malido perceived that injectable contraceptives should never be reinforced, as they exposed young women to potential for other STIs or secondary HIV transmissions.

> *"Condoms are safe in preventing HIV transmission; with injectable contraceptives, adolescents are at risk of STIs, in addition to many effects of injectable contraceptives like delayed fertility return or bleeding, which if coupled with HIV positive status can likely expose them to anaemia."* (Mrs Tayenda, Mwatitha's service provider, 39)

When attempting to understand their social realities, most adolescents revealed they were depressed and choose the image in Fig 4. 'Am depressed'. They perceived that they were not being loved and accepted within the family and the society, which resulted in a loss of sense of belonging and adult support. Loss of parental love and adult support networks, made the majority engage in sexual relationships with older partners. This exposed them to gender power inequalities, increased their vulnerability to sexual exploitation and early childbearing as they succumbed to unsafe sexual practices to maintain their sources of support, social status, and identity.

**Enabling an understanding of ambiguities and complex issues.** Among our key interview questions was the issue of status disclosure particularly, with whom did the adolescents talk to about their HIV positive status and why they preferred talking to the chosen people and not others about their status. For the caregivers, we asked them, to whom could they disclose the positive HIV status of their adolescents. In response to this question, Fatsani's mother indicated that though status disclosure to significant others was necessary, she felt that disclosure

of the youth's status simultaneously potentiates parental status, thus subjecting their family to social isolation and embarrassment. On the other hand, she felt disclosing status to sexual partners was important to prevent the risk of transmitting the virus and legal implications.

> *". . ...we said let's not tell them (the children) about the status . . . . . . . . ...will they not tell others? Will that not disclose our status? If others know, how will they treat us as a family? Later we disclosed the status to them because the older daughter (Fatsani) kept on demanding reasons for taking her daily drugs; but we told them not to tell our relations, family friends to avoid social isolation. But I have always told her to disclose the status to her sexual partners for fear of being sued if she does not disclose her status and infect her partners. Personally, I feel she better gets someone who is already infected so that they disclose their status to each other without any challenge. . . ..." (Mrs Ndazi, Fatsani's caregiver, 41)*

While the caregiver's (Fatsani's caregiver) concerns seemed to revolve around legal implications if the partners discovered themselves that her daughter was HIV positive, the adolescent (Fatsani) related her position on non-disclosure to a previous relationship termination by her first sexual partner after disclosing her status. In revealing image choices regarding individuals she would talk with about her positive HIV status, Fatsani's account presented similar images of the constraints set by her mother, and extended the secrecy to her sexual partners contrary to her mother's opinions. Compared to other adolescents in our sample, in her response to the question, Fatsani restricted her choice of images to services providers and family members (Fig 5: 'Family members' and Fig 6: 'Service providers').

The stigma associated with HIV and previous termination of her relationship after status disclosure made Fatsani keep her status a secret even to her sexual partner. Fatsani's choice of images confirmed silence and secrecy as coping strategies against the social stigma and demonstrated the need to protect her family from social embarrassment as well as future termination of relationships with sexual partners. However, secrets concerning status interfered with optimal relationships exposing Fatsani to subsequent emotional adjustments and risk of secondary HIV transmission or re-infection as reflected below:

> *You may not understand the pain of having this shameful disease; you could not even tell your boyfriend for fear of having the relationship terminated, but will he not know about it? How will he react? The pain of feeling lonely, not loved again, no! My mum, sisters and nurses have to know because they support me. It seems even mum and dad are ashamed of this disease, they wouldn't want to hear that I have told someone but a boyfriend; I can't! Why should I? It's a pin code (secret) I need a companion.* (Fatsani, 17)

The contributions of the young women and their support affirmed that living with HIV involves keeping secrets for various reasons at different levels. The consequences of this silence were not fully explored but, tangentially, we heard about risks of secondary HIV transmission, loss of social networks, or deprivation of intimate relationships, all of which were linked to affecting their ability to live positively.

**Gaining insight into perceived inter-generational dissonance.** 'My story' book helped to generate and enable rich interview discussions on type and approach of health services, sexual behaviours and experiences. It gave female adolescents control over the representation of their own social realities as they chose images that best matched their individual sexual experiences and needs. Although service providers reported that the adolescents could not open up on their sexual experiences and needs despite prolonged involvements, the use of 'my story' book enabled them to share such information. In the case of Mwatitha responding to '*what helps*

*you cope up with living with your condition*?' She chose images in Figs 2, 5 and 6, on 'having a sexual partner' 'family members' 'service providers' respectively and Fig 7 on 'Type and approach of health services' to explain her coping strategies.

During interviews she explained that:

*. . .you may not understand why I engaged in a sexual relationship at the age of 12. They (service providers including caregivers) could not be happy if they had heard that I was doing it (sex) but as I was growing, I felt lonely and not loved in my aunt's house, I needed someone to love me; my boyfriend was always there for me. I had to do it (sex) as a token of appreciation for his support. . . . . .. I ensured that they (my service providers and aunt) were not aware of our relationship; they advise us (adolescents living with HIV) to abstain for fear of transmitting the virus; but how could I with all the love and support I was getting from him? At clinic, I could not ask for condoms for I did not want them (service providers) to note that I was doing it (sex). . . . (Mwatitha, 18)*

Both Fatsani and Mwatitha opened their communication with '*you may not understand*' which highlighted the perceived inter-generational dissonance, but 'my story' book vectored their expression of experiences and needs in a way that transects diagnosis, generation, and power roles. For example, Mwatitha presented the rationale and promotion of abstinence amongst adolescents living with HIV by service providers and their caregivers in an attempt to mitigate risk of secondary HIV transmission to their sexual partners or re-infection. This was confirmed by the following sentiments:

*"I was shocked to discover that she (Mwatitha) has a boyfriend, I expected that she would abstain especially considering her positive status. Will she not pass on the virus to her boyfriend? Surprisingly, when I visited the centre. . . . . . . . .I heard one of the girls almost of her age on a mobile phone, I believed she was talking to her boyfriend, 'please take care do not move out with other girls, I love you.' I was paralysed, these girls are engaging in sexual relationships; we expect the impossibilities, they are engaging in sex as they are growing. . . . . ."* (Mrs Metani, Mwatitha's caregiver, 40)

*"I feel stigma and discrimination she (Tawina) encounters affects her access to condoms; most providers know that she is HIV infected because of her skin condition. So if they see her collecting condoms it could be an issue; the negative attitudes—they would not expect her to be sexually active; they would think she is being suicidal to sexual partners. This makes young people prefer buying condoms from shops than collecting them from the centre; personally, I don't mind offering her condoms, but if she is seen by other service providers, will question her, why does she want to kill many? Reflecting on challenges with condom use in these rural areas, they would prefer she abstains—this discourages young women from using SRH services."* (Mr Pamba, Tawina's service provider, 25)

Service providers asserted that when they considered the ages and the small stature of some of the adolescents, they could not think they could be sexually active, only to be shocked later seeing some of them pregnant, contrary to their expectation that the young women were abstaining from sex.

## Discussion

Images in research whether researcher or participants generated are not a taken-for-granted record of the subjects' everyday life [6]. They are a representation, a version of events [27] co-

produced by the researcher and the study participants subject to interpretation by both of them, as it is with any other source of data [28]. Images also do not become data until layered with interpretation and analysis [19]. In the current study the 'my story' book comprising of images and sentence completion exercise enabled adolescents to discuss their SRH needs and experiences, how they engaged with SRH services including aspects of their lives that might be difficult or sensitive to express in words to an adult researcher (See S1 Dataset: Study Conclusion). This satiates the findings of Fraser [6], who indicated that it can be difficult for an adult researcher to collect data on sensitive issues from young people during research. 'My story' book generated insights over and beyond the conventional data collection techniques developed for adults [27, 29]. The use of 'my story' book therefore positioned female adolescents as 'experts' of their own lives, capable of discussing their sexual experiences with an adult researcher without being coerced.

Creatively using images and sentence completion as data collection methods enabled female adolescents to express their own real life issues which were contrary to service providers' perspectives particularly on sexual experiences. Teachman and Gibson [18] argued that dominant groups assert power through dogmatic, authoritative truth claims which could be contrary to social world of the interviewed. While abstinence could be an idealized behaviour for the adolescents from the service providers' and caregivers' perspectives [5], adolescents indicated that conforming to sexual abstinence could not resonate with the economic imperative and the shared social identity which served as the basis for the receipt of effective support from peers or sexual partners [4, 30]. Through 'my story' book, some adolescents spoke of the disregard for safer sexual behaviours by their cohorts because of their search for social, economic, and psychological well-being which they could not through interviews alone. They pretended to be 'doing the right thing' so as not to be seen as sexually active by caregivers and the service providers; they covertly engage in such relationships as conveyed through use of 'my story' book. It is in this context that O'Connell [31] refutes the premise that 'children have nothing of interest or importance to tell researchers about their lives, and that adults understand much better than them on what is good for them and how events impact on them'. The 'my story' book enabled the adolescents' stories to be captured through images, metaphors, ensuring the researcher, service providers understood their socio-economic and sexual realities in diverse ways, and making the ordinary become extraordinary. The 'my story' book embraces O'Connell's [31] conjecture that adolescents be regarded as social actors with the capacity to construct and negotiate their personal and social contexts. This research revealed and gave voice to adolescents' emotions, needs, and challenges which then became relatable to other sources of information and evidence, which is a critical component of 'my story' book's methodological contribution.

With the use of 'my story' book, we found that chosen images provided a useful record of the phenomenon under study that could be mulled over, repeatedly viewed and reflected upon as the analysis was in progress. Focus on the number of images chosen, the rationale and meaning ascribed to the chosen images, and how the images impacted the range and depth of data collected overall, helped the researcher to make a more robust contribution to the body of knowledge. For instance, the use of images and the sentence completion exercise revealed that male relationships seemed to positively influence the young woman's self-image and self-worth, which contradicted service providers' expectations of abstinence among adolescents living with HIV. Our methodological choice to use images and sentence completion therefore empowered the adolescents, by asking them to perform an action they were skilled at and comfortable with, to reveal their own secrets and share their sensitive stories.

There are a number of limitations with the use of 'my story' book in the current study. The plan was for the adolescents to complete 'my story' book at their own time and pace within the

scheduled period of completing the 'my story' book (two months), in order to give them a better sense of control over the activity. However, some adolescents took long than planned period (more than two months) to undertake the exercise, and complete 'my story' book as they were making every effort in seeking a safe place and to maintain privacy and confidentiality of the issues written in 'my story' book. For future studies adopting use of 'my story' book with young people, plans are needed in advance on how adolescents could be assisted in terms of locating and accessing space to conceal their books, thus when and where to complete 'my story' book and where to store 'my story' book during the period of completing the exercise to ensure maximum privacy and confidentiality. Whilst the study included only adolescents who were well and had a cognitive capacity to complete 'my story' book, it could have been significant as well to empower participants who were unwell to share their stories through 'my story' book hence boosting their self-esteem. Participants whose caregivers and service providers refused to assemble a complete 'case' might have had significant issues or challenges to influence practice and policy. However, 'my story' book as a methodological approach can be used beyond the group of female adolescents whose cases were analysed (to all young people in different settings).

## Conclusion

Critically reflecting on the rich and sensitive data that was collected from the adolescents through the 'my story' book, it is evident that the book could be considered as one of the innovative visual data collection methods in exploring life experiences and needs of the female adolescents including sensitive and personal issues which could not easily been expressed with words only. In this context, the use of 'my story' book assisted the adolescents to actively construct the story of their experiences, attach meaning to the chosen images which communicated not only key events but also contexts, feelings, values, challenges and opinions as they were explaining the experiences and needs in reference to the chosen images. It also resulted in a shift towards a more equitable power distribution, providing the autonomy needed for female adolescents to reveal personal stories about their sexual and reproductive needs contrary to what caregivers and service providers believed as the truth about adolescents' sexual experiences.

## Supporting information

**S1 Dataset.**
(PDF)

## Acknowledgments

We are very grateful to all female adolescents who gave up their time to complete 'my story' book. We would also like to thank all caregivers and service providers working at Lighthouse and Baylor College of Medicine in Malawi for their support throughout data collection period. It is important to note that a portion of 'my story' book was previously published (in brief) under data collection.

## Author Contributions

**Conceptualization:** Gertrude Mwalabu, Ida Mbendera.

**Formal analysis:** Gertrude Mwalabu.

**Investigation:** Gertrude Mwalabu.

**Methodology:** Gertrude Mwalabu.

**Writing – original draft:** Gertrude Mwalabu.

**Writing – review & editing:** Gertrude Mwalabu, Ida Mbendera, Pammla Petrucka, Violet Manjanja.

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
