## [Decision Letter · Decision Letter 0]

7 Sep 2020

PONE-D-19-34742

FEMALE ADOLESCENTS LIVING WITH HIV TELLING THEIR STORY THROUGH ‘MY STORY’ BOOK IN MALAWI: A VISUAL METHODOLOGY INNOVATION

PLOS ONE

Dear Dr. Mwalabu:

Thank you for submitting your manuscript to PLOS ONE. After careful consideration, we feel that it has merit but does not fully meet PLOS ONE’s publication criteria as it currently stands. Therefore, we invite you to submit a revised version of the manuscript that addresses the points raised during the review process.

In your resubmission, ensure that the document is free from any typos or grammatical errors and check to ensure consistency of the style throughout the document, particularly as it relates to PLOS One requirements.

We look forward to receiving your revised manuscript.

Kind regards,

Mary C Smith Fawzi, ScD

Academic Editor

PLOS ONE

2. We note that Figures [1,3,4,5,6,7,8] includes an image of a [patient / participant / in the study]. 

Reviewers' comments:

Reviewer's Responses to Questions

**Comments to the Author**

1. Is the manuscript technically sound, and do the data support the conclusions?

Reviewer #1: Partly

Reviewer #2: Yes

2. Has the statistical analysis been performed appropriately and rigorously? 

Reviewer #1: No

Reviewer #2: N/A

3. Have the authors made all data underlying the findings in their manuscript fully available?

Reviewer #1: Yes

Reviewer #2: Yes

4. Is the manuscript presented in an intelligible fashion and written in standard English?

Reviewer #1: No

Reviewer #2: Yes

5. Review Comments to the Author

Reviewer #1: This is an interesting area. However, the readability of the manuscript is obstructed by counteracting or inconsistent information, unstructured presented methodology and lack of succinctness in the presentation of information. The introduction needs heightening to clearly establish the problem statement. At the same time, the discussion should fall in line with the objective of the study.

Abstract

• Background information in this section should be revised to match that in the body’s introduction. This section is informing about the RSH challenges adolescents to encounter while the main document’s introduction is discussing communication and the method of data collection -my storybook

• Under the findings section, were these themes a representation of data from all sources?

Introduction

• In general, the introduction needs to be tuned to strongly establish the problem statement to support the objective. The authors should consider highlighting the essentialness of trying/testing ‘my story’ book data collection method on this vulnerable population.

• Also considering that this is a relatively novel method of data collection, readers (I am) would be interested to learn more about it, as it has been briefly introduced. What is it? What is already known? Who is it appropriate for? How is it used? Is it better than other methods? Has it been used before…..where and on who? Etc.

• A little background on adolescent girls with HIV in Malawi would also add value

Methods

This section needs reconstruction and rearrangement of information to improve readability and cohesion. There is a lot of areas in the methods section that have been presented in a backward forward manner or repetitions. Could the authors revise and summarise the information in this section to come up with a reasonably succinct and -still- informative section? I commend you to read articles published by Plose one for guidance.

Recruitment and sampling

• This section has information beyond recruitment and sampling only, I would suggest removing the title or include subsections, which can also help with the organisation of the information.

• It is not clear why caregivers and service providers were recruited in a study that aimed at exploring how a ‘my story’ book approach was developed and used as the basis for visual depictions and in-depth interviews with a group of female adolescents living with HIV’. Again, these carers shared experienced on care provision, which is contrary to the objective of the study……... Could the authors justify or rectify this, please

• How many multidisciplinary centres were samples drawn from? And where were they located- providing the name of the town (s) would be helpful

• The minimum duration of clinic attendance is stated as 6 months and participants unaware of their statuses were excluded. How did the authors balance these, considering that young people can attend a clinic for years, only to learn their status the very day for recruitment?

• On inclusion criteria, could the authors clarify what unwell and too sick meant in their research experience? Also, could they elaborate on how this was redefined throughout the study duration since participants kept these booklets at home for two months

• The second paragraph under the recruitment section is not related to the context. This information is better suited in the introduction section. At the same time, the study is aiming at enhancing data collection method because adolescents are usually shy/uncomfortable with the extant commonly used approaches to data collection. This paragraph is discussing sexual risk behaviours and SRH needs, which is good, but it has to be rephrased be relatable to the study content in the introduction and also to strengthening the problem statement. i.e. linking the identified needs/challenges to background, research and innovative data collection methods. The last sentence is highlighting biasness/inequality would the authors consider removing it?

• It is stated that ‘all 15 to 19 year old female adolescents who met the selection criteria were invited to participate’ on page 3, and on page 5 they state that ‘out of 20 who met the inclusion criteria…..’ Could the authors confirm that 20 represent the overall numbers of those who met inclusion criteria in all centres. Also, why did the recruitment begin from 16 years of age than the planned age of 15?

• Page 5: information beginning from ‘Out of the 14 participants, ten were from urban settings and four from rural locations should move to the results section.

o Were strategies to diversify sample used when recruiting these participants? Please include a statement to support the recruitment of more urban than rural participants and, less young youths ie 18 and 19-year-old?

• Please clarify why ethical approval was sought from a UK based ethics committee for a study done in Malawi.

• Ethics information under methods should be briefly stated as a better explanation is given at the end of the document

• Who conducted the interviews? How were the booklets returned? Was there an interview guide?

• No information is given on interviews of parents and guardians.

• Page 6 paragraph one stated that evidence-informed the choice of images, please include the themes that directed the choice……what message did the selected pictures carry

• Page 6 last paragraph, first two sentences starting with ‘The images were used as a catalyst’ should blend with paragraph 3 which is starting with ‘In this study, the adolescents were invited to put stickers on images’…….before proceeding to interviews

• Examples of stickers and generated information on paged 7 and 8 should be summarised please, or move some information to findings section

• Page 8 last paragraph, most information is a repeat of what has already been stated earlier

• In data coding section, it should come out clearly who was the researcher and the research supervisors…..Would use the initial of the authors. Include their credentials/research experience to/and justify their qualitative research abilities

• The researchers do indicate that ‘Codes, sub-themes, and themes were identified from individual accounts within the case then conclusions were dawn’ but it is unclear how this was done. Please provide a clearer explanation of steps taken to analyse data

• How was the data from storybook triangulated with that from in-depth interviews

• Was data saturation considered?

• Some elements of rigor have been highlighted under data analysis, which is good. Would this be made clearer to elaborated what components of trustworthiness were achieved by what action? This could come on a separate subsection.

• On table 2 , two last themes appear to discuss similar content and might as well be merged. Are these themes from interviews? If yes, please include in the caption

• Was there some translation involved, how was this carried out? By who? What was translated?

• Please use a COREQ reporting checklist for your methods

Findings

• The authors have exhibited their identified themes in table 2 that are not discussed in the findings section. Also, two sets of themes are presented, which is pretty confusing. Please explain the role of these themes in this study by justifying why Creating meanings, Confirmatory and complementary evidence and Power relations in inter-generational research are discussed instead.

• Contradicting statement on Page 13 ‘Among our key research questions was the issue of status disclosure’,

Discussion

The discussion section should be revised to analyse matters relatable to the objective of the study. The why, the how and the so what/what next of this innovative process of data collection are lacking. How were conclusions for the feasibility of my storybook data collection procedures proved ‘acceptable’ through this study? This needs to be established in the discussion section.

• The objective of this study as highlighted in the introduction section is different from ‘In seeking to examine female adolescents’ sexual experiences and needs we explored how female adolescents engage with sexual and reproductive health services and whether the HIV services meet the varying needs of this group as they grow into adulthood’

• Paragraph 2 is a stand-alone paragraph. It is not linked to my story book

• Only discuss findings that are presented in the results section ie findings from caregiver are not included in the findings section

Limitations

Omit the study limitation section/ merge relatable limitations only

Also revise and include limitations that are associated with conducting a qualitative study and interpreting the data thereof.

Consider tackling on disparities of your sample characteristic while aligning with the recommendation to use this approach to counsel (youth?) on one on one basis

Conclusion

Could the conclusion be shortened and without references, please?

Reviewer #2: This is a timely manuscript that addresses innovative, youth-friendly approaches with adolescents and openly discusses their complexities and sexuality. I have made the following observations and comments:

1. There are repetitions in manuscript which can be catered for if there is a stand-alone paragraph on 'what' and 'how' the innovative visual qualitative approach, the 'my story' book has been used at global, regional levels and particularly Malawi. This will fit best as part of the background and in the objectives

a. The paragraph or two may be put in the background, before the methods section

2. Please confirm the ages of the participants. The abstract says 16-19 years, while the methods section refers to 15-19 years.

3. The authors should confirm demographic information for caregivers and service providers.

a. It is also not clear how these were sampled and recruited.

4. The authors might need to consider to move the paragraph on the rationale of the study to the background section to situate the gap, objectives etc.

5. There is a need for a stand-alone paragraph about authors talking about their positionality. There is a need to acknowledge the power-related issues that come with the role of older woman/nurse/midwife working with adolescents living with HIV.

6. Please specify what 'other means of survival' entails.

7. There is a need for the authors to dedicate a paragraph on recruitment of adolescents, their caregivers, health workers and the advisory committee, 2) data collection processes using 'my story' book should be laid out, i.e. using images, stickers, and completion of 'my storybook'; including the use of in-depth interviews, the key examples of questions presented to adolescents/

a. The authors may consider presenting their research questions rather than figure 1 and 2 as they tend to read more like results.

8. Please provide the names of districts, 'multi-disciplinary centres', health facilities providing HIV care

9. Please specify the free websites or public domains that were accessed and their URL for the images presented in the manuscript

a. Specify if the authors' used researcher generated images or internet-based.?

10. Include the paragraph on the advisory committee under data analysis

11. There is a need for the authors to reduce the repetitions on the importance of using the 'my story' book approach'. It will help to make it succinct and coherent

12. There are a few grammatical errors in the results section, discussion sections

6. PLOS authors have the option to publish the peer review history of their article (what does this mean?). If published, this will include your full peer review and any attached files.

Reviewer #1: No

Reviewer #2: **Yes: **Blessings N. Kaunda-Khangamwa

---

## [Author Response · Author response to Decision Letter 0]

4 Nov 2020

Responses to reviewers 1 and 2 have been attached as separate documents.

---

## [Editor Report · Decision Letter 1]

26 Nov 2020

PONE-D-19-34742R1

FEMALE ADOLESCENTS LIVING WITH HIV TELLING THEIR STORY THROUGH ‘MY STORY’ BOOK IN MALAWI: A VISUAL METHODOLOGY INNOVATION

PLOS ONE

Dear Dr. Mwalabu,

Thank you for submitting your manuscript to PLOS ONE. After careful consideration, we feel that it has merit but does not fully meet PLOS ONE’s publication criteria as it currently stands. Therefore, we invite you to submit a revised version of the manuscript that addresses the points raised during the review process.

As the academic editor for your manuscript, I have some additional changes that you need to make before the manuscript can be considered for publication. Please refer to my list of changes below and resubmit your manuscript in addition to a letter that indicates how you addressed each of these points.

A rebuttal letter that responds to each point raised by the academic editor and reviewer(s). You should upload this letter as a separate file labeled 'Response to Reviewers'.A marked-up copy of your manuscript that highlights changes made to the revised version. You should upload this as a separate file labeled 'Revised Manuscript with Track Changes'. These track changes should be new tracks on the clean copy that you created for your resubmission.An unmarked version of your revised paper without tracked changes. You should upload this as a separate file labeled 'Manuscript'.

We look forward to receiving your revised manuscript.

Kind regards,

Mary C Smith Fawzi, ScD

Academic Editor

PLOS ONE

Additional Editor Comments (if provided):

Title page:

-For the listing of authors remove the period after Malawi (in two places)

-For the correspondence section state: “… is a University Lecturer, Dean of Faculty of Nursing Studies at Kamuzu College of Nursing, University of Malawi, a consultant, and an expert in Adolescent Health, Sexual and Reproductive Health (SRH), Gender Issues, Human Immunodeficiency Virus (HIV), and Acquired Immune Deficiency Syndrome (AIDS) and conducts research with young people based in Malawi.”

Abstract page:

-Replace the word ‘Background’ with ‘Introduction’.

-Last sentence of introduction: remove words ‘gain voice’.

-First sentence of methods: replace ‘living with HIV share’ with ‘living with HIV to share’.

-Second sentence of methods: replace ‘researcher appreciate’ with ‘researcher to appreciate’.

-Last sentence of conclusion: replace ‘one-to-one’ with ‘one-on-one’.

Background:

-Instead of using ‘Background’ refer to this section as the ‘Introduction’.

-First paragraph: spell out ‘SRH’ the first time that you use it.

-Third sentence of first paragraph: Break down into 2 sentences since this is a long sentence.

-Middle of first paragraph: replace ‘grow to adulthood’ with ‘transition to adulthood’.

-End of first paragraph: replace ‘articulate the realities of lives including’ with ‘in articulating the realities of their lives, including’.

-Second sentence of second paragraph: replace ‘that becomes difficult to be completely bridged.’ With ‘and it can become difficult to bridge this gap.’

-Middle of second paragraph: replace ‘mature female researcher’ with ‘adult female researcher’.

-Middle of second paragraph: replace ‘perceive that what they’ with ‘ensure that what they’.

-Middle of second paragraph: replace ‘influence policy’ with ‘as well as influence policy’.

-Middle of third paragraph: replaces ‘However, my story’ book’ with ‘However, ‘my story’ book’.

-First sentence of fourth paragraph: replace ‘formed interview’ with ‘formed the interview’.

-Final paragraph of Introduction: replace ‘ensuring the researcher’ with ‘ensuring that the researcher’

-Very end of final paragraph of the introduction: revised to ensure that reader is absolutely clear that this paper is about the ‘my story’ book methodology and not about the content of the interviews. You can do this by starting the last sentence of the introduction to say: ‘The goal of this paper is to…’

Methods:

-Change title of this section to ‘Materials and Methods’.

-Remove any mention of the names of clinical sites to protect confidentiality. You should refer to the size of the patient population, scope of services (e.g. only pediatrics? Only HIV?). Refer to urban versus rural sites (e.g. 2 urban and 1 rural site).

-Under ‘recruitment and sampling’: replace ‘imposed to reduce any perceived additional’ to ‘implemented to reduce any additional’.

-Under ‘recruitment and sampling’: replace ‘Out of the 20 adolescents, who’ with ‘Out of the 20 adolescents who’.

-Under ‘recruitment and sampling’ specify out of the 14 the number who are from rural versus urban areas.

-For consent indicate that everyone 18 and above provided written informed consent (no need to include verbal). For those under 18 indicate that the parent or legal guardian provided consent and the adolescent study participant provided assent (in this order).

-Under ‘recruitment and sampling’: remove “(as outlined in PLOS consent form)”.

-Under ‘recruitment and sampling’ (last paragraph of this section: replace ‘they operate in’ with ‘they operate’.

-Last paragraph, recruitment and sampling: rephrase where it states ‘mature lady’ and include term adult. This last section should be rephrased as follows: ‘Based on her own experience as an adult female and mother, the researcher felt that older adolescents had a wider experience than younger adolescents, consistent with existing literature [provide at least 3 references]. Therefore, the researcher decided to include older participants, particularly ages 15-19 based on the age range she felt the young women could have broader experiences and be open to discuss issues like sexual behaviours.’

-Under ‘data collection using ‘my story’ book’: replace ‘formed ‘sentence’ with ‘formed the ‘sentence’.

-Under ‘data collection using ‘my story’ book’: It is unclear to the reader what sections of the story book are included as part of an interview with the researcher and what part they fill in by themselves (if that occurs at all)?

-Under ‘data collection using ‘my story’ book’: Where it states the majority of adolescents filled all sections of the story book, indicate the actual number.

-Under ‘data collection using ‘my story’ book’: Is there any data that is not interview data (e.g. the participant just ‘filled it out’?). If yes, please indicate how this data was included, e.g. was it typed into files to be included in the qualitative analysis?

Under ‘data collection using ‘my story’ book’, last paragraph: replace ‘interviewing process’ with ‘interview process’.

-Data analysis section: should be limited to 3-4 paragraphs maximum (do not include bullet points in this section).

-Data analysis, first paragraph: replace ‘researchers view’ with ‘researchers to view’.

-Data analysis: remove Table 1 and just state that the results related to the content are published elsewhere (and state reference) or that they are forthcoming (if not yet published).

-For methods: make it clear to reader how data were collected for caregivers and health care providers. For example, if there are in-depth interview guides state that and there is audio-recording, etc., provide this detail, and so forth.

Results:

-First sentence of first paragraph: replace ‘Through sentence’ with ‘Through the sentence’.

-Under ‘creating meanings’: it is not clear if the loss of parental affection is related to death of a parent?  Try to make this more clear.

-Under ‘creating meanings’: replace ‘were viewed as competent’ with ‘were competent’.

-Under ‘creating meanings’: for the quote that says ‘I got connected to him’ remove the extra space.

-Under ‘creating meanings’: for paragraph starting with the word ‘Similarly’, remove the space before this word.

-Under ‘creating meanings’: replace ‘In contrary, majority’ with ‘In contrast, the majority’.

Discussion:

-paragraph before limitations: replace ‘images, how did the images impacted on the range and depth of data collected overall, helped’ with ‘images, and how the images impacted the range and depth of data collected overall, which helped’.

Limitations:

-Remove this title and just start this paragraph with: ‘There are a number of limitations in the current study’.

-Remove ‘(taking into account their shorter attention spans)’.

-Refer to possible implications of not including participants who were too sick or unwell (e.g. implications for empowerment?)

-Refer to possible implications of only including adolescents in which a caregiver and health care provider could also be recruited.

Overall:

-References in the text are not in Vancouver format. Instead of XX et al., 2019, you should indicate the number in brackets, e.g. [5].

-References at end should all be consistent and in Vancouver format and in one size font.

---

## [Author Response · Author response to Decision Letter 1]

23 Mar 2021

Figure files have been uploaded to the Preflight Analysis and Conversion Engine (PACE) digital diagnostic tool, https://pacev2.apexcovantage.com/.

---

## [Decision Letter · Decision Letter 2]

7 Jun 2021

PONE-D-19-34742R2

FEMALE ADOLESCENTS LIVING WITH HIV TELLING THEIR STORY THROUGH ‘MY STORY’ BOOK IN MALAWI: A VISUAL METHODOLOGY INNOVATION

PLOS ONE

Dear Dr. Mwalabu,

Thank you for submitting your manuscript to PLOS ONE. After careful consideration, we feel that it has merit but does not fully meet PLOS ONE’s publication criteria as it currently stands. Therefore, we invite you to submit a revised version of the manuscript that addresses the points raised during the review process.

The manuscript has been further evaluated by two reviewers, and their comments are available below.

The reviewers have raised a number of minor concerns regarding the manuscript’s clarity and detail. They specifically request changes to improve organization and more well-defined terminology, in addition to providing more supporting information in the Introduction.

Could you please carefully revise the manuscript to address all comments raised?

We look forward to receiving your revised manuscript.

Kind regards,

Avanti Dey, PhD

Staff Editor

PLOS ONE

Journal Requirements:

Reviewers' comments:

Reviewer's Responses to Questions

**Comments to the Author**

1. If the authors have adequately addressed your comments raised in a previous round of review and you feel that this manuscript is now acceptable for publication, you may indicate that here to bypass the “Comments to the Author” section, enter your conflict of interest statement in the “Confidential to Editor” section, and submit your "Accept" recommendation.

Reviewer #1: (No Response)

Reviewer #2: (No Response)

2. Is the manuscript technically sound, and do the data support the conclusions?

Reviewer #1: Partly

Reviewer #2: Partly

3. Has the statistical analysis been performed appropriately and rigorously? 

Reviewer #1: N/A

Reviewer #2: N/A

4. Have the authors made all data underlying the findings in their manuscript fully available?

Reviewer #1: Yes

Reviewer #2: Yes

5. Is the manuscript presented in an intelligible fashion and written in standard English?

Reviewer #1: No

Reviewer #2: No

6. Review Comments to the Author

Reviewer #1: The authors have made a lot of helpful revisions and much improvement is seen in the manuscript as a result. There remain, however, some issues with the methods and the results sections that need rectifying. Lastly, the length of the manuscript can be reduced by thorough editing, which can also enhance the readability and cohesion in some areas.

1. Ethics statement

a. I can see that assents for adolescents under 18 were obtained. I am wondering if parental informed consent (not permission) was not obtained for the younger ones.

2. Introduction

a. Please include the year in the second sentence.

b. Please specify the type of support for adolescents in HIV programs. This should also link to the succeeding sentence for cohesion i.e. who benefits from these services as far as age is concerned (since you are talking about transition)? Is there any evidence that highlight the current status of issues surrounding transitioning into adult care? Consider the articles from Malawi that discuss health services eg content in number 2 and 4 of your bibliography and this article- “Uptake of health services by youth living with HIV: a focused ethnography”.

c. Consider re-organising sentences in the first paragraph to support the main topic, which I suppose is the epidemiology of HIV and SRH. For instance, the sentence “Yet, young people are frequently reported to be reluctant to discuss sexual matters….” does not seem to link well with the rest in this paragraph. Likewise, the sentence on lack of knowledge on experience is not supporting the SRH needs as stated in the former sentence. Ultimately, the content does not strongly highlight the need for an innovative data collection tool. I recommend a separate paragraph that can discuss challenges encountered while accessing/providing SRH including communication hierarchies and barriers to strengthen the problem statement (this was also highlighted in the first review). This should also provide a better transition into the next paragraph, which is discussing communication techniques.

3. Considering there are 4 authors, pay attention to how you are using ‘the researcher’ throughout the document- maybe lead author?

4. Consider moving paragraph 3, 4, 5, 6; which are more of procedures than an introduction, into the methods section.

5. How many experts were consulted?

Methods

6. Consider looking into the information provided under on Ethics statement and Ethics approval and consent to participate.

a. Was parental consent sought for adolescents below 18 years of age? Please clarify.

7. Under recruitment and sampling:

a. Consider substituting the second sentence with a statement on how and where the service providers were recruited from.

b. On inclusion criteria: did literacy matter in this study?

c. Consider merging the first part of paragraph 1 with paragraph 2 (discussing recruitment), and inclusion criteria should come in a separate paragraph,- for cohesion

d. Consider moving findings in paragraph 3 and 4 into the results section e.g. Six reported living with their biological parent(s) (father/mother), three were married e.t.c.

e. I feel working as a nurse/midwife or affiliation with an HIV management centre does not affect the relationship with both genders- please specify the meaning here. This paragraph can also be summarised and shortened.

f. In general, there is a lot of information that is not suitable for this section. Consider relocating such information. This section is also long, it can be reduced to 2 paragraphs at most.

g. The authors should consider highlighting reasons why health workers and parents were included in a study that was examining data collection methods on youth- you have talked about triangulation later in the document, maybe this should come earlier?

8. Data collection

a. Please clarify which transcript was audited (the original or the translated)?

b. How was translation bias managed considering only one person was involved in the process?

c. This fact is repeated ‘The researcher transcribed all the recorded interviews verbatim soon after each interview.”

9. Data analysis

a. The authors have mentioned that codes, subthemes and themes were identified but the procedures undertaken to analyze data are not clearly coming out for reproducibility. Could the authors make this section clearer? (This was also recommended in the previous review).

b. How was the data handled considering it came from different sources? Where/when did the merging/triangulation of findings occur? (This was also highlighted in the previous review)

10. Results

a. The second paragraph is not clear, consider rephrasing.

b. Please maintain consistency with the wording in the themes

c. The theme confirmatory and complementary is not quite relatable to the content, which is mostly on disclosure of HIV status and associated prejudice. It this the confirmation of adolescents' perspectives of disclosure of HIV with their caregiver’s views? If this is the case, I recommend renaming the theme since all other themes have confirmatory information from multiple sources.

d. ‘Among our key research questions was the issue of status disclosure particularly’ contradicts the aim of the study. Did you mean interview questions? (this was also queried in the last review)

e. Consider moving the last sentence of theme two to the discussion section

f. Is it ‘power relations in inter-generational research’ or ‘enabling intergenerational research?’

g. Similarly, the first two sentences under the last theme are more suitable in the discussion section information of disclosure from caregiver

h. Again, the title for theme three (research) does not relate to the content (perceived disagreement). Consider renaming the theme. Please add quotes from caregivers and health workers to support the claim of inter-generational dissonance.

11. Discussion

a. Please cite the first sentence

b. Please avoid repeating phrases or sentences e.g. it can be methodologically difficult for an adult researcher to find techniques to listen to the young people’s voices in research

c. The discussion section is much improved and reads well.

Reviewer #2: I would like to commend the authors for thoroughly revising the paper. However, the paper cannot be accepted for publication in the current form. There are minor changes to be considered and a few areas that need to be strengthened for coherence.

1. There are a few paragraphs where the authors repeat their ideas.

a. Pages 9 and 19 when the authors refer to ‘safe places’.

b. Pages 9 and 10 as they talk about ‘keeping the thematic log.’

2. The paragraphs in the introduction need to be strengthened.

I appreciate the arguments from Kellett, Chambers and the appropriateness of using the ‘my story book’.

a. There is a need to know more about the concept of ‘my story book’ or the ideas of ‘memory book’.

b. How did the ‘book’ work in the past? How they have been used global, sub-Saharan Africa and Malawi are still missing.

c. ‘How?; When? and where? These concepts have been used, will strengthen the gap that Teachman and colleagues and Wills and colleagues presented.

d. As it is, the paragraphs from pages 4-7.

i. The concept of my story books…

ii. The main themes of my story books…

iii. The discussion held with the researcher…

The above paragraphs leave the reader hanging and need real-life supporting evidence from the region or elsewhere. In addition, they continue to read like a paragraph that is part of the methodology section rather than the introduction. We need a solid and convincing introduction section.

3. The methodology is still too long and it shares some results too.

a. On page 7, the authors refer to 14 adolescents who completed the story book.

b. The researchers/author’s positionality should be reflected as part of data collection rather than recruitment and sampling.

4. The results section.

a. The first paragraph can be dedicated to demographic information and the emerging themes or contribution to knowledge.

b. Rather than reflecting on what the paper highlights, please rephrase and talk about key themes.

c. The case of Tawina may follow Ziliwe or Zaiwa’s narratives to maintain coherence.

5. Discussion section

a. Please include tangible evidence from the region (sub-Saharan Africa) on how using memory books/my story book or ‘visual research methods’ are better methods.

b. The comparative discussion should not be between quantitative vs qualitative methods but participatory or visual methodologies.

7. PLOS authors have the option to publish the peer review history of their article (what does this mean?). If published, this will include your full peer review and any attached files.

Reviewer #1: **Yes: **Maggie Zgambo

Reviewer #2: **Yes: **Blessings N. Kaunda-Khangamwa

---

## [Author Response · Author response to Decision Letter 2]

22 Jul 2021

The reviewers suggestions have been implemented in the manuscript and were very helpful in regard to improvement of the manuscript

---

## [Decision Letter · Decision Letter 3]

25 Aug 2021

FEMALE ADOLESCENTS LIVING WITH HIV TELLING THEIR STORY THROUGH ‘MY STORY’ BOOK IN MALAWI: A VISUAL METHODOLOGY INNOVATION

PONE-D-19-34742R3

Dear Dr. Mwalabu,

We’re pleased to inform you that your manuscript has been judged scientifically suitable for publication and will be formally accepted for publication once it meets all outstanding technical requirements.

Kind regards,

Tanya Doherty, PhD

Academic Editor

PLOS ONE

Additional Editor Comments (optional):

Please address the final comments from one reviewer in terms of amendments to your description of recruitment, data coding and analysis.

Reviewers' comments:

Reviewer's Responses to Questions

**Comments to the Author**

1. If the authors have adequately addressed your comments raised in a previous round of review and you feel that this manuscript is now acceptable for publication, you may indicate that here to bypass the “Comments to the Author” section, enter your conflict of interest statement in the “Confidential to Editor” section, and submit your "Accept" recommendation.

Reviewer #1: (No Response)

Reviewer #2: All comments have been addressed

2. Is the manuscript technically sound, and do the data support the conclusions?

Reviewer #1: Yes

Reviewer #2: Yes

3. Has the statistical analysis been performed appropriately and rigorously? 

Reviewer #1: N/A

Reviewer #2: N/A

4. Have the authors made all data underlying the findings in their manuscript fully available?

Reviewer #1: Yes

Reviewer #2: Yes

5. Is the manuscript presented in an intelligible fashion and written in standard English?

Reviewer #1: No

Reviewer #2: Yes

6. Review Comments to the Author

Reviewer #1: Could the authors edit information on recruitment - page 6 as 14 participants were recruited ---- “Through purposive sampling, eight adolescents aged 15-19 were recruited from each HIV management centre, and four from the rural facility (a total of 20)”

Could information on data coding and analysis be rearranged i.e. relocating coding information in paragraph starting with trustworthiness to the paragraph on top which is discussing coding as well?

Check references.

Reviewer #2: The authors have addressed most of the issues raised, responded with richer examples across the countries. The paper is ready for editing.

7. PLOS authors have the option to publish the peer review history of their article (what does this mean?). If published, this will include your full peer review and any attached files.

Reviewer #1: **Yes: **Maggie Zgambo

Reviewer #2: **Yes: **Blessings N. Kaunda-Khangamwa

---

## [Editor Report · Acceptance letter]

7 Oct 2021

PONE-D-19-34742R3 

FEMALE ADOLESCENTS LIVING WITH HIV TELLING THEIR STORY THROUGH ‘MY STORY’ BOOK IN MALAWI: A VISUAL METHODOLOGY INNOVATION 

Dear Dr. Mwalabu:

I'm pleased to inform you that your manuscript has been deemed suitable for publication in PLOS ONE. Congratulations! Your manuscript is now with our production department. 

Kind regards, 

on behalf of

Professor Tanya Doherty 

Academic Editor

PLOS ONE